

# Real time retrieval of volcanic cloud particles and SO₂ by satellite using an improved simplified approach

**S. Pugnaghi[1], L. Guerrieri[1], S. Corradini[2] and L. Merucci[2]**

[1]{Dipartimento di Scienze Chimiche e Geologiche, Università di Modena e Reggio Emilia, 41125 Modena, Italy}

[2]{Istituto Nazionale di Geofisica e Vulcanologia, 00143 Roma, Italy}

Correspondence to: S. Pugnaghi (sergio.pugnaghi@unimore.it)

## Abstract

Volcanic Plume Removal (VPR) is a procedure developed to retrieve the ash optical depth, effective radius and mass, and sulphur dioxide mass contained in a tropospheric volcanic cloud from the thermal radiance at 8.7, 11, and 12 μm. It is based on an estimation of a virtual image representing what the sensor would have seen in a multispectral thermal image if the volcanic cloud were not present. Ash and sulphur dioxide were retrieved by the first version of the VPR using a very simple atmospheric model that ignored the layer above the volcanic cloud. This new version takes into account the layer of atmosphere above the cloud as well as thermal radiance scattering along the line of sight of the sensor. In addition to improved results, the new version also offers easier and faster preliminary preparation and includes other types of volcanic particles. As in the previous version, a set of parameters regarding the volcanic area, particle types, and sensor are required to run the procedure. However, in the new version, only the mean plume temperature is required as input data. It this work a set of parameters have been computed for different types of plume particles (andesite, obsidian, pumice, ice, water, and sulphuric acid droplets), for both the Mt. Etna (Italy) and Eyjafjallajökull (Iceland) volcanoes, and for the MODIS Terra and Aqua instruments. Two different synthetic images, one for Mt. Etna and one for Eyjafjallajökull, are used to compare the results from the new and old procedures. Finally, a sensitivity analysis was conducted to investigate variations in VPR ash and sulphur dioxide retrievals as a function of plume altitude and particle type.





## 1. Introduction

The large volumes of ash mixed with various gases that can be released into the atmosphere during explosive volcanic eruptions sometimes form clouds that travel great distances from the source over long periods, carried by the wind. These ash clouds can be generated at any time from the eruption of any one of more than 1,200 active volcanoes scattered over the Earth's surface (Prata, 2009) and pose a real threat to air safety (Casadevall, 1994).

An effective global monitoring system today depends on the use of satellite data to detect and monitor the evolution of volcanic ash clouds. Timely information on the location, size, height, and ash content of potentially hazardous eruption clouds derived from satellite data are generated and used by the Volcanic Ash Advisory Centres (VAACs) to mitigate this type of threat and improve aviation safety (Francis et al., 2012).

Satellite sensors operating in the thermal infrared range are particularly effective for this purpose, when the interaction of volcanic ash with electromagnetic radiation makes it possible to detect and monitor volcanic clouds even at night. The algorithms developed exploit in various ways the reverse absorption of the brightness temperature observable in the channels centred at 10.8 and 12 microns. This feature is used both for discriminating ash and meteorological clouds (Prata 1989a, 1989b), and for quantifying the mass, optical thickness, and effective radius of the ash contained in volcanic clouds (Wen and Rose, 1994).

Several algorithms were developed in the early efforts to detect volcanic clouds and retrieve the ash and $SO_2$ contents, as discussed in a recent critical review (Clarisse and Prata, 2016). Among the new algorithms the simplified approach of the VPR is distinguished by its ease of use and speed of calculation, making it highly effective for monitoring. Another advantage of the VPR approach is that it only requires the plume temperature as additional input, providing fresh estimates of ash and $SO_2$ as soon as new satellite images of an ongoing eruption become available (Pugnaghi et al., 2013; Guerrieri et al., 2015).

The VPR procedure was developed using thermal infrared (TIR) data collected by the Moderate Resolution Imaging Spectroradiometer (MODIS) instrument on board the Terra and Aqua polar platforms, and by the Spinning Enhanced Visible and Infra Red Imager radiometer (SEVIRI) on board meteorological satellites positioned on MSG geostationary orbits.

This paper aims to present the VPR procedure in an improved and simplified form as developed for the selected case studies of the Mt. Etna (Italy) and Eyjafjallajökull (Iceland) eruptions. Section 2 is dedicated to a theoretical description of the novel improvements of the VPR procedure, while section 3 presents and discusses the results obtained for the validation case studies. Section 4 provides



conclusions. Further theoretical details are included in Appendix A, while the VPR coefficients are
tabulated in Supplement for different types of plume particles (andesite, obsidian, pumice, ice, water,
and sulphuric acid droplets), for both the Mt. Etna and Eyjafjallajökull volcanoes, and for the MODIS
Terra and Aqua instruments.
**2. Theory**
The Volcanic Plume Removal (VPR) procedure (Pugnaghi et al. 2013; Guerrieri et al. 2015) is a
linearization of the radiative transfer equation developed to retrieve, from multispectral satellite
images at 8.7, 11, and 12 μm, the ash optical depth at 550 nm ($\delta^*$), effective radius ($R_e$), mass ($M_a$),
and sulphur dioxide mass ($M_s$) of a tropospheric volcanic cloud. The parameters required to apply
the VPR are specific for a given volcano, type of plume particles, and sensor on board the satellite
and these are easily determined *a priori* using the MODTRAN radiative transfer model. Once they
have been computed, the only additional inputs required are the multispectral image and the mean
plume temperature.
Figure 1 shows the VPR procedure flowchart (dashed rectangle). The land-sea mask is usually
available with the radiance data while the operator has to define the plume mask and possibly the
meteorological cloud mask. For the multispectral sensors the plume mask can be derived from ash
detection techniques based on Brightness Temperature Difference (BTD, see Prata, 1989b) and
successive improvements (see Millington et al., 2012, Pavolonis et al., 2013), principle components
analysis (Hillger and Clark 2002a,b), or neural networks (Picchiani et al., 2014). The volcanic cloud
temperature input data can be obtained from VIS/TIR ground-based cameras (Scollo et al., 2014),
ground radar (Montopoli et al., 2014; Marzano et al., 2006; Corradini et al., 2015), lidar system
(Scollo et al., 2012) measurements, or from multispectral satellite data using different techniques like
*dark pixels* (Prata and Grant, 2001; Corradini et al., 2010), $CO_2$ slicing (Menzel et al., 1983; Platnick
et al., 2003), $H_2O$ intercept method (Nieman et al., 1993), tracking of volcanic cloud centre of mass
(Guerrieri et al., 2015), or inversion schemes based on Optimal Estimation (Francis et al., 2012).
The first step of the VPR is definition of the virtual image with the removed volcanic cloud and
computation of plume transmittances for the three bands considered (8.7, 11, and 12 μm). In the
earlier VPR approach, the atmosphere above the plume was assumed to be negligible and the results
were adjusted with a cubic relationship, derived by fitting an adequate set of MODTRAN simulations
(Pugnaghi et al. 2013; Guerrieri et al. 2015). The transmittance values at 11 and 12 μm were used to
define maps of $R_e$, $\delta^*$ and $M_a$, while the sulphur dioxide abundance map was estimated from the





transmittance at 8.7 µm. Finally, the wind speed at the plume altitude was used to reconstruct the flux
at the vents, considering both the ash mass and $SO_2$ maps (Merucci et al., 2013; Guerrieri et al., 2015;
Merucci, 2015).

4       The novel VPR procedure described here applies a new atmospheric model for estimating

volcanic cloud transmittance (white box, inside the dashed square in Fig. 1). Here both the
transmittance $\tau^{"}$ and the up-welling radiance $L_{uo}^{"}$ of the layer of atmosphere above the plume are
considered (as shown in the scheme in Fig. 2). The term representing the surface thermal radiance
scattered by the volcanic particles along the line of sight of the sensor is now also considered (not
shown in the scheme of Fig. 2).
With this atmospheric model, the plume radiance $\boldsymbol{L_p}$ measured by the sensor can be approximated by
the parabolic trend (see Appendix A for a detailed description):
$$L_p = -\alpha \cdot \tau_p^2 + \left[ L_o + \alpha - B_p \cdot \tau^{"} - L_{uo}^{"} \right] \cdot \tau_p + \left[ B_p \cdot \tau^{"} + L_{uo}^{"} \right] \qquad (1)$$
where $\alpha$ is a term mainly proportional to $\varepsilon \cdot B(T_s) \cdot \tau$; $\varepsilon$ is the surface emissivity, $B(T_s)$ is the Planck
emission at the surface temperature $T_s$, and $\tau = \tau' \cdot \tau^{"}$ is the transmittance of the whole atmosphere
($\alpha$ also depends on the aerosol optical depth, but this effect is important mainly for very optically
thick pixels); $L_o$ is the radiance at the sensor with the plume removed; $B_p$ is the Planck emission at
the mean plume temperature $T_p$; $\tau_p = \tau_a \cdot \tau_s$, is the plume transmittance where $\tau_a$ is the aerosol
transmittance, and $\tau_s$ is the part due to sulphur dioxide. From these definitions it follows that if $SO_2$
is absent then $\tau_s = 1$; and if the aerosol optical depth $\delta = 0$, then $\tau_a = 1$.
**2.1 Absence of sulphur dioxide**
If sulphur dioxide is absent or if only the thermal bands not affected by $SO_2$ are considered, in
Eq. (1) $\tau_p$ can be substituted with $\tau_a$ representing only the ash component.
Fig. 3a shows a series of MODTRAN simulated radiances at the sensor versus the plume
transmittance obtained specifically for the band at 11 µm of the MODIS-Aqua sensor, pumice (Volz,
1973) ash type, and a set of possible plume configurations (see Supplement for details).
The parameter values of the parabolic fit of the radiance $L_p$ versus the plume transmittance $\tau_a$ shown
in Fig. 3a, $L_p \cong \sum_{i=0}^{2} a_i (\tau_a)^i$, are reported in Table 1. By definition: $a_0 + a_1 + a_2 = L_0$. In this case
the sum is 8.57 (W m$^{-2}$ sr$^{-1}$ µm$^{-1}$), as can be seen in Fig. 3 for $\tau_a = 1$.



The parabolic trend shown in Fig. 3a changes according to the state of the atmosphere, the surface
characteristics and, of course, the position of the volcanic cloud, composition, and ash content.
Approximating the radiance $L_p$ expressed as a function of the plume transmittance $\tau_a$ with two linear
trends, for high radiance values (i.e. the transparent pixels of the plume) and for low values (more
opaque plume pixels), if the surface characteristics do not vary excessively over time, it can be
observed that the linear trends always intersect close to the same transmittance value (named $\tau_t$).
Figure 3b shows that $\tau_t \approx 0.3$. Clearly, the gains and offsets of these two linear trends also change
according to the state of the atmosphere, plume temperature, and so on. These two linear fits, are
characterised by four parameters. However, only the offset (named $B_{up}$) is required to fit the
transparent part because the radiance $L_0$ is known from the plume removal part of the procedure.
Similarly, if the intersection point of the two linear trends is known, the offset of the opaque part
(named $B_{dn}$) is sufficient to determine the linear fit.
Summarising, by knowing the air temperature $T_p$ at the mean plume altitude (thus $B_p$ term) and
the radiance $L_o$ with the plume removed, it is possible to estimate the aerosol plume transmittance $\tau_a$
directly from the radiance measured by the satellite (without atmospheric correction or radiative
transfer models) using Eq. (2) (red line) and, if necessary, Eq. 3 (blue line) of Fig. 3b:
$$L_p = \left[ L_o - B_{up} \right] \cdot \tau_a + B_{up} \qquad (2)$$
If the computed transmittance $\tau_a$ is lower than $\tau_t$ (intersection point), then the plume transmittance
is recomputed by:
$$L_p = \left[ (L_t - B_{dn})/\tau_t \right] \cdot \tau_a + B_{dn} \qquad (3)$$
where $L_t$ is the radiance at the sensor computed using Eq. (2) for a plume transmittance $\tau_a = \tau_t$.
Figure 4a shows that in the 11 μm band there is a linear relationships between the two aforementioned
offsets $B_{up}$, $B_{dn}$ and the Planck emission of the plume $B_p$. A similar relationship also exists for the
other two bands (obviously, for the band centred at 8.7 μm, sulphur dioxide must be absent) and for
other volcanic particle types (see Supplement). Figure 4a also shows that the plume transmittance at
the intersection point $\tau_t$ is almost constant with only a small dependence on $B_p$.
Therefore:
$$B_{up} = a_{up} \cdot B_p + b_{up} \qquad (4)$$



$$\tau_t = a_{tt} \cdot B_p + b_{tt} \qquad\qquad (5)$$
$$B_{dn} = a_{dn} \cdot B_p + b_{dn} \qquad\qquad (6)$$

### 2.2 Presence of sulphur dioxide

The presence of sulphur dioxide complicates transmittance retrieval at 8.7 µm because weak $SO_2$
absorption affects this band. If the aerosol component of the plume transmittance at 8.7 µm is known,
then the radiance at the sensor without the presence of sulphur dioxide $L_a$ can be computed using the
Eqs. (2) and (3). A knowledge of radiance due only to aerosols makes it possible to define the
following simple equation:
$$L_p = [L_a - B_s] \cdot \tau_s + B_s \qquad\qquad (7)$$
where $L_p$ is the total plume radiance measured by the sensor, $L_a$ is radiance due to the aerosol
components of the plume, and $\tau_s$ is the plume transmittance due to $SO_2$ absorption. $B_s$ is a constant
which takes into account the plume temperature, plume position, and state of the atmosphere above
the plume and it is computed using a linear function of $B_p$:
$$B_s = a_s \cdot B_p + b_s \qquad\qquad (8)$$
Figure 4b shows the trend of $B_s$ versus $B_p$ derived from a complete dataset of MODTRAN
simulations for Mt. Etna measured with a MODIS-Aqua instrument.
Therefore, to compute $\tau_s$ from Eq. (7), it is necessary to know $L_a$ which is derived from Eqs. (2) and
(3) when $\tau_a$ for the band at 8.7 µm is known. The transmittance $\tau_a$ can easily be computed for
pumice-type ash particles because a very good correlation exists between $\tau_{a,8.7}$ and $\tau_{a,11}$ (see Fig. 5a,
and Pugnaghi et al. 2013). The fit is a cubic polynomial: $\tau_{a,8.7} = \sum_{i=0}^{3} a_{i,8.7} \left(\tau_{a,11}\right)^i$ and the
parameter values from the MODIS sensors on board the Terra and Aqua satellites are reported in
Table 2. These parameters are an improved version of those reported in Pugnaghi et al. (2013),
because in the first version of the VPR the thermal radiance scattered along the line of sight of the
sensor was ignored.



Unfortunately, for other particle types (see Supplement), the correlation between $\tau_{a,8.7}$ and $\tau_{a,11}$ is
not always good, as in the example of Fig. 5b showing the scatter plot for water droplets.
Nevertheless, it should be noted that this correlation becomes very good if only particles of the same
effective radius $R_e$ are considered.
In these cases with non-pumice ash types, the aerosol transmittance $\tau_{a,8.7}$ at 8.7 μm can be obtained
from the formula:
$$\tau_{a,8.7} = e^{-\mu \cdot \delta_{8.7}} = e^{-\mu \cdot m_{8.7} \cdot \delta^*}$$    (9)
where $\mu$ is the optical air mass factor, $\delta_{8.7}$ is the vertical optical depth, $\delta^*$ is the vertical optical depth
at 550 nm, and $m_{8.7}$ is the gain of the linear relationship which gives the optical depth $\delta_{8.7}$, when $\delta^*$
is known; the gain $m_{8.7}$ is a function of the effective radius $R_e$ and is known from the MODTRAN
simulations (Guerrieri et al., 2015).
To sum up, the novel VPR procedure first computes the 11 and 12 μm band transmittances (as
indicated in the flowchart of Fig. 1), and from these the aerosol optical depth at 550 nm ($\delta^*$) and the
effective radius ($R_e$) of each pixel of the plume (Pugnaghi et al. 2013); then the aerosol transmittance
at 8.7 μm ($\tau_{a,8.7}$) is obtained using Eq. (9).
Finally, the transmittance $\tau_{s,8.7}$ (derived from Eq. 7) is used to estimate the SO$_2$ columnar abundance
$C_s$, given the proper absorption coefficient $\beta$ (Pugnaghi et al. 2013) and the optical air mass $\mu$ factor.
$$\tau_{s,8.7} = e^{-\mu \cdot \beta \cdot C_s}$$    (10)
The subsequent steps of the VPR procedure have not been changed and can be found in Pugnaghi
et al., (2013). Nevertheless, to conclude the theoretical discussion, it is important to note the
superposition effect of ash and sulphur dioxide on the radiance measured by the sensor. This means
that the proposed VPR procedure can also work well in cases of a *double-plume* at different
temperatures, for example if an ash plume is located directly above or below a sulphur dioxide plume.
**3.  Validation test cases**
To test the improved version of the VPR procedure two trial synthetic images were defined as
described in Corradini et al. (2014), for both the MODIS-Aqua effective wavelengths, depicting a
uniform ocean surface under a cloudless sky, and a plume of known spherical particles.





The first image is characterized by an atmospheric situation, ocean temperature, and the particle
type typical of the Sicilian Mt. Etna volcano, while the second is adapted to match the Icelandic
Eyjafjallajökull volcano. Figure 6 shows the two RGB colour composite synthetic images with the
radiances of the channels centred at 8.7, 11, and 12 μm respectively. The left plate shows the Mt.
Etna scenario with a plume of the same shape as the volcanic cloud detected by MODIS-Aqua during
the 26 October 2013 eruption at 12:20 GMT. The right plate shows the Eyjafjallajökull scenario,
depicting a portion of the Eyjafjallajökull plume detected by MODIS-Aqua on 11 May 2010 at 14:05
GMT.
**3.1 The Mt Etna-Pumice scenario**
The synthetic atmosphere in the Mt. Etna image in Fig. 6a is derived from the radiosonde pressure,
temperature, and humidity (PTH) profiles measured by the Trapani (western tip of Sicily, Italy) WMO
station 16429 on 26 October 2013 at 12 GMT, while the plume mask and the vertical zenith angles
used to prepare the synthetic image are derived from the actual MODIS-Aqua data collected on
October, 26, 2013 at 12:20 GMT. The plume in the synthetic image is defined as 1 km thick, located
between 7 and 8 km, containing pumice ash (Volz, 1973) and $SO_2$. It has a Gaussian shape moving
from the centre to the edge and ranging from 10 to 1 g m$^{-2}$ columnar $SO_2$ abundance, and from 1.5 to
0.1 ash optical depth $\boldsymbol{\delta^*}$ (AOD at 550 nm). Therefore, a minimal quantity of sulphur dioxide and ash
is always present in the plume, and the effective radii $R_e$ of the particles have a uniform distribution,
on a logarithmic scale, in the range 0.8-7 μm.
Table 3 shows that by excluding the $SO_2$ total mass, all the retrieval values of the new version of the
VPR are closer to the true values than the old version. Both versions estimate a lower mass of ash in
the volcanic cloud, but this probably also implies a greater burden of $SO_2$ detected with the old VPR.
The retrieval of the total ash mass computed with the new VPR is better not only because it is closer
to the true value, but also because both the estimated effective radius and optical depth used in mass
estimation are closer to the true values.
Figure 7 shows the scatter plots of $R_e$, $\delta^*$, $M_a$, and $M_s$ versus the true values (synthetic image). All
the scatter plots show a widening dispersion with increasing values. Fig. 8 reports (on the left) the
trends of $R_e$ and $\delta^*$ mean values retrieved with VPR using different input plume altitudes, and on the
right the trends of ash and $SO_2$ total mass. As described in Pugnaghi et al. (2013), Guerrieri et al.
(2015), and Merucci (2015), the effective radius and aerosol optical depth at 550 nm are derived from
the transmittances retrieved at 11 and 12 μm, and then the ash mass is computed in each pixel with



the Wen and Rose (1994) simplified formula. The trend of $R_e$ versus volcanic cloud altitude is almost
flat, while the optical depth $\delta^*$ shows a clear drop with height. The best retrieval (closest to true
values) is at 7 km rather than the height used of 7.5 km. This is also true for $SO_2$ total mass.
Figure 9 shows the VPR retrievals for the Mt. Etna scenario giving as input the right plume
temperature and all the types of particles reported in Tables S1, S2, S3, S5, S6, S7 (see Supplement).
The upper plates show the mean effective radius ($R_e$, left) and the mean optical depth ($\delta^*$, right). The
lower plates show the retrievals of ash (left) and $SO_2$ (right) total mass. Among the different types of
ash, andesite gives the worst effective radius and optical depth results, with respect to the true values.
Nevertheless, because the two retrieved variables $R_e$ and $\delta^*$ compensate each other, all the ash types
considered give good estimations of the ash total mass. Conversely, for ice and water the results
retrieved for the total mass are much higher and divergent from true values. Finally, by varying the
ash type, the total ash mass exhibits a much lower variability when compared to that of $SO_2$.
**3.2 The Eyjafjallajökull-Andesite scenario**
The second synthetic image shown in Fig. 6b was created considering the state of the atmosphere
derived from the PTH vertical profiles measured at Keflavik (WMO station 04018) on 11 May 2010
at 12:00 GMT. On that day, MODIS-Aqua captured the Eyjafjallajökull eruption during its transit at
14:05 GMT. As in the previous Mt. Etna eruption, the plume mask and the vertical zenith angles used
in the synthetic image derive from the actual MODIS-Aqua data. The plume was again 1 km thick,
but located lower than the previous case at between 4 and 5 km, containing spherical particles of
andesite (Pollack et al. 1973) and $SO_2$. The same ranges and distributions of $SO_2$ columnar content,
ash optical depth $\delta^*$ at 550 nm, and effective radius $R_e$ were used, as for Mt. Etna.
Once again Table 4 demonstrates that the new version of the VPR generates better estimations
compared to the old one, with all the parameters exhibiting difference percentages lower than 10%.
The older version detects the presence of ash in a smaller number of pixels, but its greater effective
radius and optical depth partly compensate for the smaller number of detected ash plume pixels in the
final mass estimation.
The numbers of detected pixels for the sulphur dioxide component are in close agreement with the
true value in both versions, and they retrieve a similar total mass of $SO_2$.
Figure 10 shows the scatter plots of $R_e$, $\delta^*$, $M_a$, and $M_s$ versus the true values. The correlation is quite
good up to about $R_e = 5$ μm, $\delta^* = 1$, $M_a = 10\ gm^{-2}$, and $M_s = 7\ gm^{-2}$; wider dispersions can be
observed for higher values.




Fig. 11 shows the trends of the VPR $R_e$ and $\delta^*$ mean values (left), and total $M_a$ and $M_s$ (right)
retrieved as functions of the input plume altitude. $R_e$ and $\delta^*$ retrievals exhibit opposite trends which
compensates each other, making the final retrieval of total ash mass less sensitive to plume altitude.
In fact, from 3 to 5 km the ash mass ranges between 11 and 15 kt with a true value of about 13 kt.
Only the optical depth at the right plume altitude (4.5 km) is very close to the true value. The best
effective radius is at a mean plume altitude greater than 5 km and the best total mass is at about 4 km.
However, the result obtained appears to be the best compromise. A greater plume height (e.g. 5 km)
would mean a better $R_e$ but a worse $\delta^*$ and also a worse total mass. Conversely, a lower plume height
(e.g. 4 km) yields to very good total mass, but worse $R_e$ and $\delta^*$ values.
Finally, as for Mt. Etna, the VPR results were considered using as input the actual plume temperature
and all the types of particles reported in Supplement, including the Eyjafjallajökull ash type (Peters,
2013, referred in the figures as Eyja ash). In Fig. 12 the upper plates show the mean effective radius
$R_e$ (left) and optical depth $\delta^*$ (right) versus the different types of particles; the lower plates show the
retrievals of ash and $SO_2$ total mass.
Only andesite and Eyjafjallajökull ash types give good results in both mean effective radius and
optical depth, while obsidian is reasonably good only for the first parameter. The ash total mass
exhibits quite constant values between 10 to 15 kt for all the four ash types, close to the true value of
13 kt. Vice versa, the total mass values retrieved for ice and water droplets were much higher and
very different from the true values. As noted in the previous Etna scenario, their reciprocal difference
is mainly due to different radius thresholds used in the procedure for water droplets and ice. The
performance of sulphur dioxide total mass retrieval is seen to be strongly affected by the type of
particle used in the procedure. Only andesite gives a good result in this case. Finally, sulphuric acid
retrieval performance is different from both ash and the water-ice pair.
**4.  Conclusions**
The new version of the VPR presented here is an approximated procedure. It uses only the mean
altitude cloud temperature as input to directly interpret MODIS-TIR multispectral images and retrieve
particle effective radius, optical depth, mass of the particle utilized, and mass of sulphur dioxide
contained in each pixel of the volcanic cloud. The VPR approach requires no atmospheric correction
because this is implicit in the procedure itself. The retrieval of effective radius, optical depth, and
sulphur dioxide abundance is derived from estimation of plume transmittances in the bands centred
at 8.7, 11, and 12 μm. This article presented a novel and effective improvement in the transmittance



estimation scheme. In the new VPR, plume transmittance is obtained from the radiance measured by
the sensor using two simple linear relationships; one for the most transparent part of the plume and
one for the most opaque. These two linear trends account for two minor terms not considered in the
previous version: the layer of atmosphere above the plume and the thermal radiance scattered along
the line of sight of the sensor. Approximation for very thick/opaque volcanic clouds (transmittances
lower than 0.05) is less effective. The improvement only involves the computation of volcanic cloud
transmittance, and no other parts of the previous procedure have been modified. Nevertheless, the
improvement has a dual positive effect: 1) it is simpler to use and provides more accurate results than
before; 2) the preliminary work to compute the parameters required by the procedure (the parameters
reported in Supplement) is easier than before and requires less processing time. The new VPR
procedure was assessed against the older version by applying it to synthetic images generated using
two real examples from the Mt. Etna (Italy) and Eyjafjallajökull (Iceland) volcanoes. The percentage
difference between the average input data of the synthetic images and the mean results of the
improved VPR ranges between 2-13 % for both Mt. Etna and Eyjafjallajökull, while the old VPR
produced ranges between 4-68 % (see Tables 3 and 4), confirming the improved performance of the
new version.
The correlation coefficient between the transmittance of the volcanic cloud simulated by MODTRAN
and the corresponding transmittance retrieved by VPR is reported in the last column of tables S1 to
S7, in nearly all cases this being close to one. However, the mean percentage errors of the retrieved
effective radius, optical depth, ash mass, and sulphur dioxide mass expected in a real example may
be greater than those reported in Tables 3 and 4. This is because the two synthetic images considered
here exhibit a uniform and perfectly clear sky, a uniform ocean surface, and a volcanic cloud
comprised of known spherical ash particles.
**Appendix A**

26        The radiance at the sensor when the volcanic cloud is absent is (see Fig. 2):

$L_o = [\varepsilon\, B(T_s) + (1-\varepsilon)\, L_d] \cdot \tau + L_{uo}$             (A1)





where: $\varepsilon$ is the surface emissivity; $B(T_s)$ is the Planck function at the surface temperature $T_s$; $L_d$ is
the atmospheric down-welling radiance; $\tau$ is the total atmospheric transmittance; $L_{uo}$ is the total
atmospheric up-welling radiance.
$\tau = \tau' \cdot \tau''$                                                  (A2)
$L_{uo} = L'_{uo} \cdot \tau'' + L''_{uo}$                                          (A3)

8         The radiance at the sensor when the volcanic cloud is present is:

$L_p = [\varepsilon\, B(T_s) + (1 - \varepsilon)\, L_d] \cdot \tau \cdot \tau_p + L_u + S$                  (A4)
where: $\tau_p$ is the volcanic cloud transmittance; $L_u$ is the current atmospheric up-welling radiance; $S$ is
the term accounting for the scattering of thermal radiance along the line of sight of the sensor.
Since surface emissivity is close to 1 (particularly above the ocean), the change of $L_d$ in the presence
of a volcanic cloud was ignored.
The atmospheric up-welling radiance in the presence of a volcanic cloud is:
$L_u = L'_{uo} \cdot \tau_p \cdot \tau'' + L_{up} \cdot \tau'' + L''_{uo}$                           (A5)
and, assuming $L_{up} = B_p \cdot (1 - \tau_p)$ where $B_p$ is the Planck function at temperature $T_p$ (air
temperature at the mean plume altitude):
$L_u = \{L_{uo} - [B_p \cdot \tau'' + L''_{uo}]\} \cdot \tau_p + [B_p \cdot \tau'' + L''_{uo}]$           (A6)





Indicating with $P$ a degree of probability of the thermal radiation being scattered along the line of
sight of the sensor, the scattering term was modelled as:
$$S = \left\{ \int_{\tau_p}^{1} \left[ \varepsilon\, B(T_s) \cdot \tau' \cdot \tau_p \right] \cdot P \, d\tau'_p \right\} \cdot \tau'' = \alpha \cdot \tau_p \left( 1 - \tau_p \right) \qquad \text{(A7)}$$
where: $\alpha = \varepsilon\, B(T_s) \cdot \tau \cdot P$.
Here $P$ is assumed to be constant even if it is a function of the ash/particle characteristics and
therefore of $\tau_p$ itself. Clearly $P = 0$ if $\tau_p = 1$.
Inserting Eqs. (A6) and (A7) in (A4):
$$L_p = \left[ \varepsilon\, B(T_s) + (1 - \varepsilon)\, L_d \right] \cdot \tau \cdot \tau_p + \left\{ L_{uo} - \left[ B_p\, \tau'' + L''_{uo} \right] \right\} \cdot \tau_p + \left[ B_p\, \tau'' + L''_{uo} \right] + \alpha \cdot \tau_p \left( 1 - \tau_p \right)$$
$$\text{(A8)}$$
Finally, recalling Eq. (A1), Eq. (1) is obtained.
**Acknowledgements**
This work was partially funded by the European Union's Seventh Framework Programme (FP7/2007-
2013) through the project APho-RISM (Advanced PRocedures for volcanIc and Seismic Monitoring)
under grant agreement number 606738.
The authors would like to thank Gavin Taylor for carefully reading the original manuscript and for
providing language corrections.



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



1    Table 1: Parameter values of the parabolic fit shown in Fig. 3a.

| $(-\alpha)$ | $\left(L_o + \alpha - B_p \cdot \tau^{"} - L_{uo}^{"}\right)$ | $\left(B_p \cdot \tau^{"} + L_{uo}^{"}\right)$ |
|---|---|---|
| $a_2$ | $a_1$ | $a_0$ |
| -1.58 | 6.92 | 3.23 |



1    Table 2: Polynomial coefficients to compute $\tau_{a,8.7}$ from $\tau_{a,11}$ for the pumice (Volz, 1973) ash type.

| Satellite | $a_{3,8.7}$ | $a_{2,8.7}$ | $a_{1,8.7}$ | $a_{0,8.7}$ |
|-----------|-------------|-------------|-------------|-------------|
| *Terra* | 0.1645 | -0.4249 | 1.2559 | 0.0050 |
| *Aqua* | 0.1412 | -0.3483 | 1.2028 | 0.0046 |





1   Table 3: Main characteristics of synthetic image, indicated as "True", together with the results of the

2   VPR procedure, both new and old versions. The air temperature at 7.5 km was used as input for the

3   VPR. The percentage differences are shown in brackets.

| Mt. Etna – Pumice; 23 October 2013<br><br>plume altitude 7-8 km | True | VPR<br><br>new | VPR<br><br>old |
|---|---|---|---|
| Mean $R_e$ (µm)<br><br>(% difference) | 2.85 | 2.92<br><br>(2.5) | 4.80<br><br>(68.4) |
| Mean $\delta^*$<br><br>(% difference) | 0.25 | 0.22<br><br>(-12) | 0.19<br><br>(-24) |
| $R_e < 2\ \mu m$ Fine particles (%) | 42 | 42 | 21 |
| $2\ \mu m < R_e < 5\ \mu m$ Mean particles (%) | 42 | 46 | 42 |
| $R_e > 5\ \mu m$ Coarse particles (%) | 16 | 12 | 37 |
| Ash mass (t)<br><br>(% difference) | 8336 | 7812<br><br>(-6.3) | 7166<br><br>(-14.0) |
| Pixels detected with ash | 7533 | 7533 | 7317 |
| $SO_2$ mass (t)<br><br>(% difference) | 19636 | 17146<br><br>(-12.7) | 18880<br><br>(-3.9) |
| Pixels detected with $SO_2$ | 7533 | 7533 | 7533 |



1  Table 4: Main characteristics of synthetic image, indicated as "True", together with the results of the

2  VPR procedure, both new and old versions. The air temperature at 4.5 km was used as input for the

3  VPR. The percentage differences are shown in brackets.

| Eyjafjallajökull - Andesite 11 May 2010 14:05; plume altitude 4-5 km | True | VPR new | VPR old |
|---|---|---|---|
| Mean $R_e$ (μm) (% difference) | 2.83 | 2.62 (-7.4) | 3.3 (16.6) |
| Mean $\delta^*$ (% difference) | 0.28 | 0.29 (+3.6) | 0.39 (39.3) |
| $R_e < 2\ \mu m$ Fine particles (%) | 43 | 46 | 38 |
| $2\ \mu m < R_e < 5\ \mu m$ Mean particles (%) | 42 | 44 | 46 |
| $R_e > 5\ \mu m$ Coarse particles (%) | 15 | 10 | 16 |
| Ash mass (t) (% difference) | 13227 | 12006 (-9.2) | 9674 (-26.9) |
| Pixels detected with ash | 10624 | 10624 | 6532 |
| $SO_2$ mass (t) (% difference) | 30724 | 28714 (-6.5) | 28235 (-8.1) |
| Pixels detected with $SO_2$ | 10624 | 10624 | 9827 |



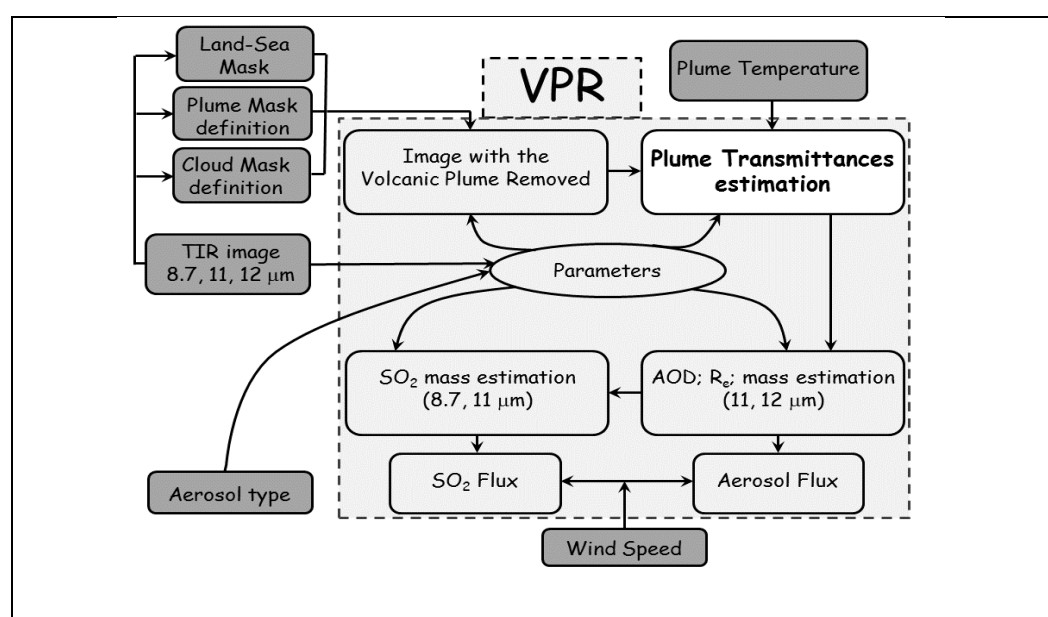

2    Figure 1. Flowchart illustrating the main steps of the VPR procedure.


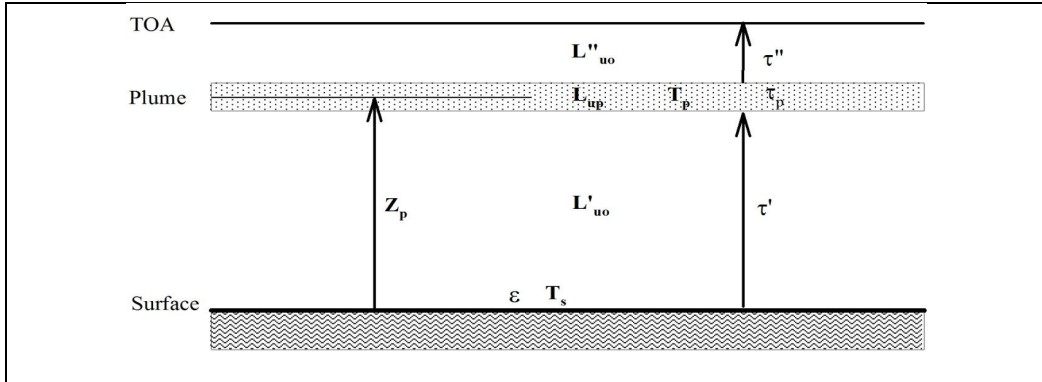

1    Figure 2. Scheme of the atmospheric model used in the improved VPR.




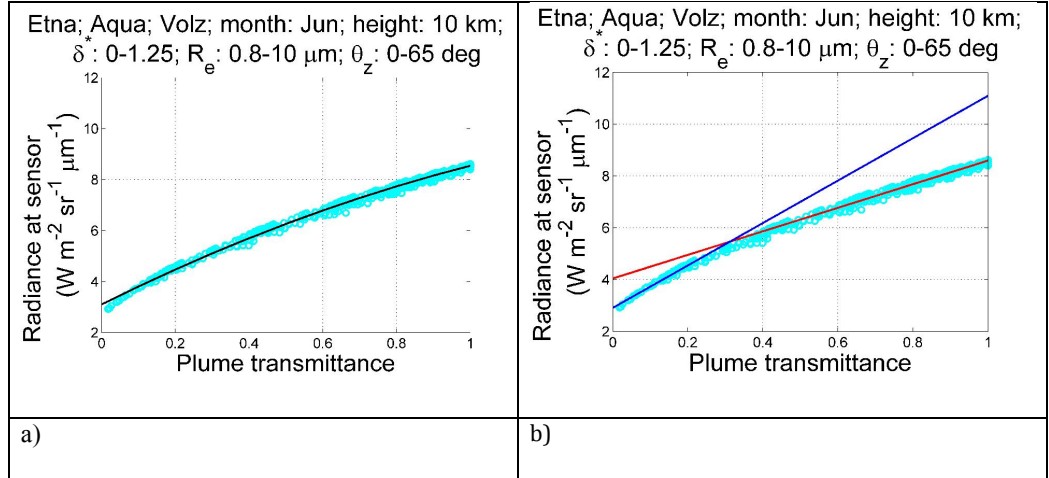

a)   b)

1   Figure 3. a) Radiances at the sensor (11 µm) vs. plume transmittances and their parabolic fit (black

2   line); b) Same radiances with the two linear fits of Eq. (2) for the more transparent part of the plume

3   (upper fit, red line), and Eq. (3) for the most opaque part of the plume (lower fit, blue line).





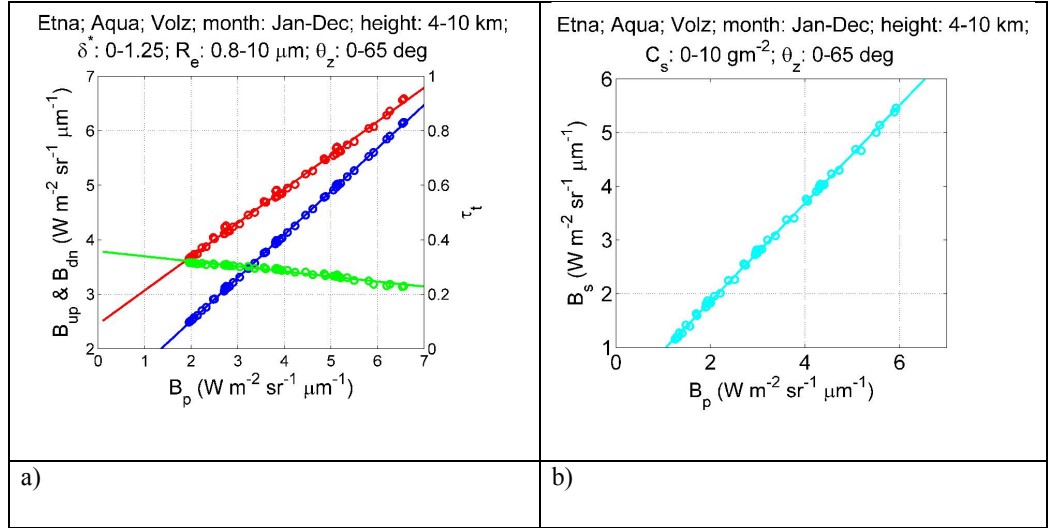

a)

b)

1    Figure 4. Linear trends of $B_{up}$ (red), $B_{dn}$ (blue), $\tau_t$ (green), and $B_s$ (cyan) versus $\boldsymbol{B_p}$ for 48 different

2    plumes (12 months and 4 heights) each obtained from a set of MODTRAN simulations.





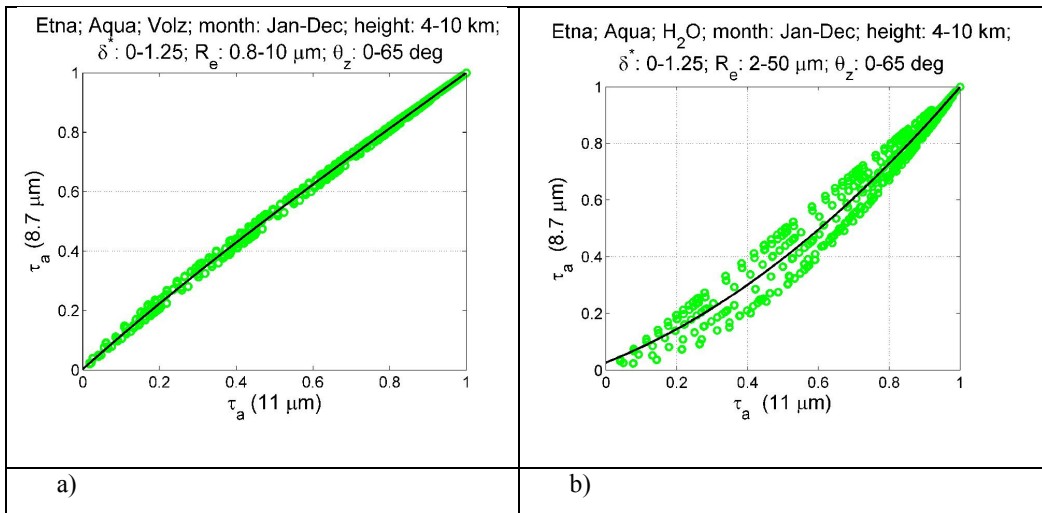

1    Figure 5. Scatter plots between the plume transmittance (obtained from a wide set of MODTRAN

2    simulations) for MODIS-Aqua bands at 11 and 8.7 μm for the pumice (Volz, 1973) ash type (a), and

3    for water droplets (b).



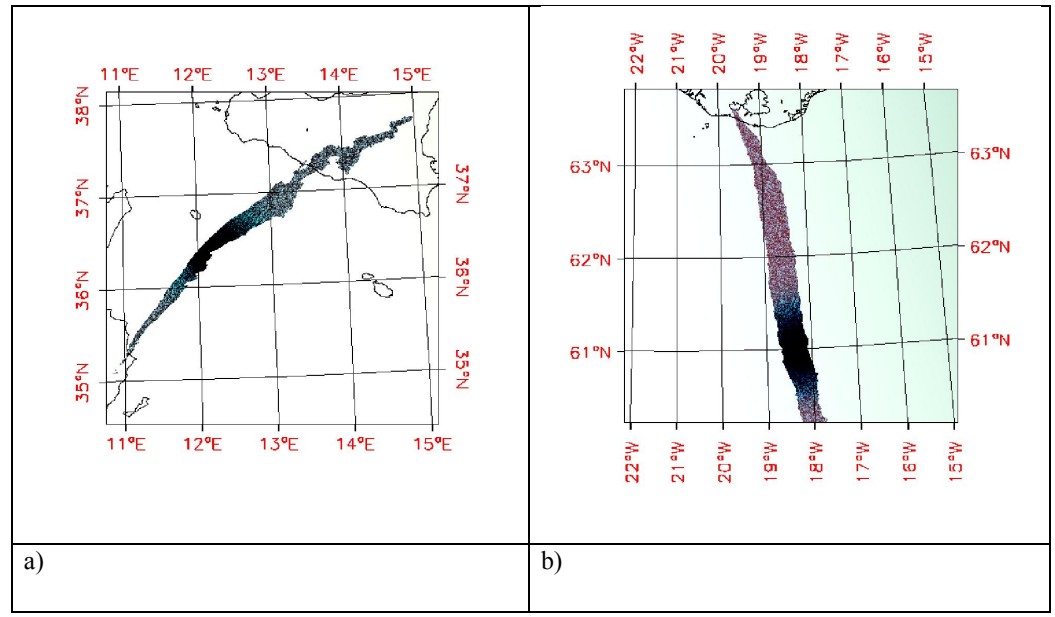

1    Figure 6. Synthetic images (radiance at the sensor); RGB: bands at 8.7, 11, and 12 µm respectively.

2    a) Mt. Etna 26 October 2013 at 12:20 GMT; b) Eyjafjallajökull 11 May 2010 at 14:05 GMT



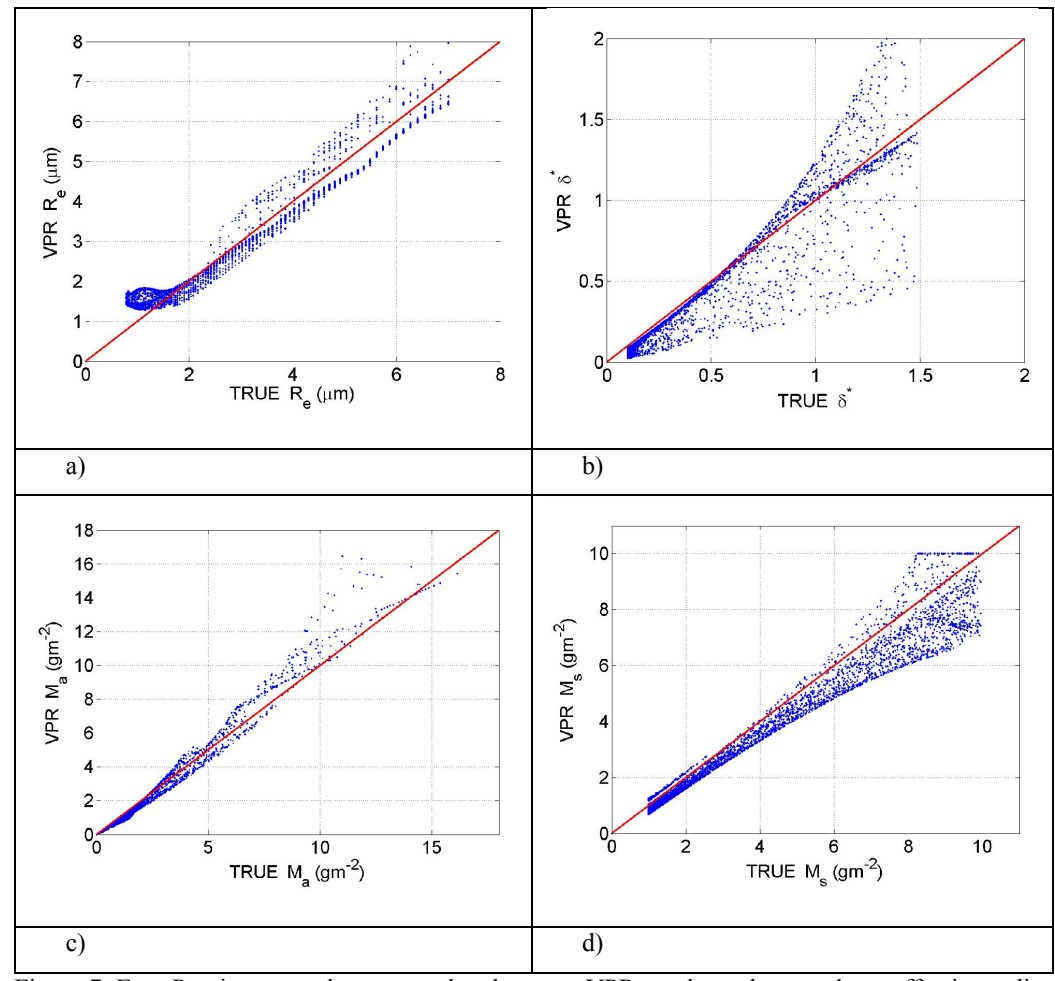

1    Figure 7. Etna-Pumice example: scatter plots between VPR results and true values: effective radius

2    (a), ash optical depth at 550 nm (b), ash mass (c), and $SO_2$ mass (d). Red line is the bisector.





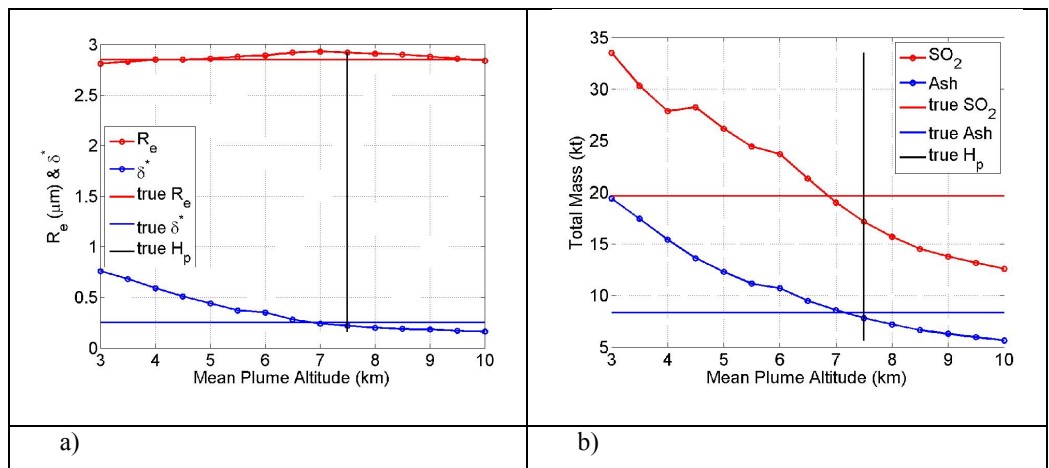

1    Figure 8. Etna-Pumice example: trends of $R_e$ and $\delta^*$ mean values (a), and ash and $SO_2$ total mass (b)

2    retrieved by VPR with different input plume altitudes.





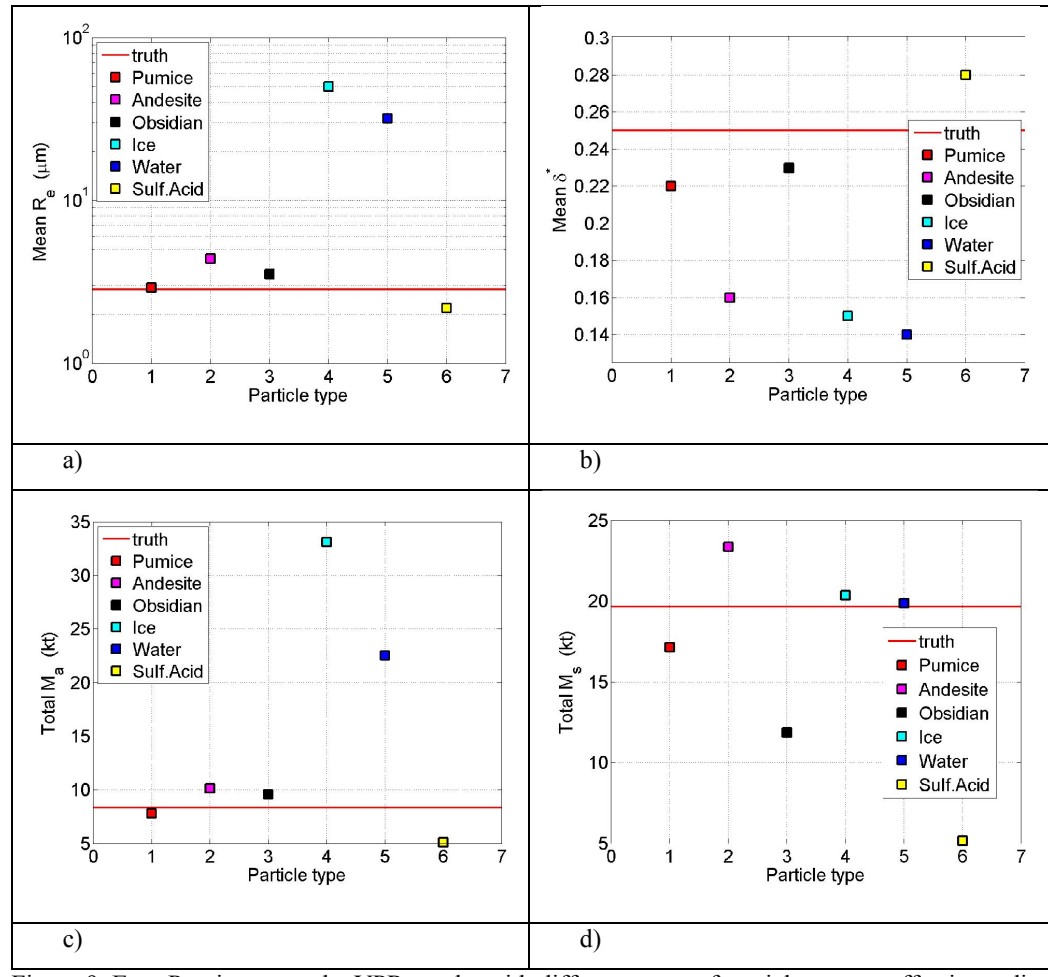

1  Figure 9. Etna-Pumice example. VPR results with different types of particles: mean effective radius

2  (a), mean optical depth at 550 nm (b), total mass (c), and total $SO_2$ mass (d). The red lines are the true

3  values.





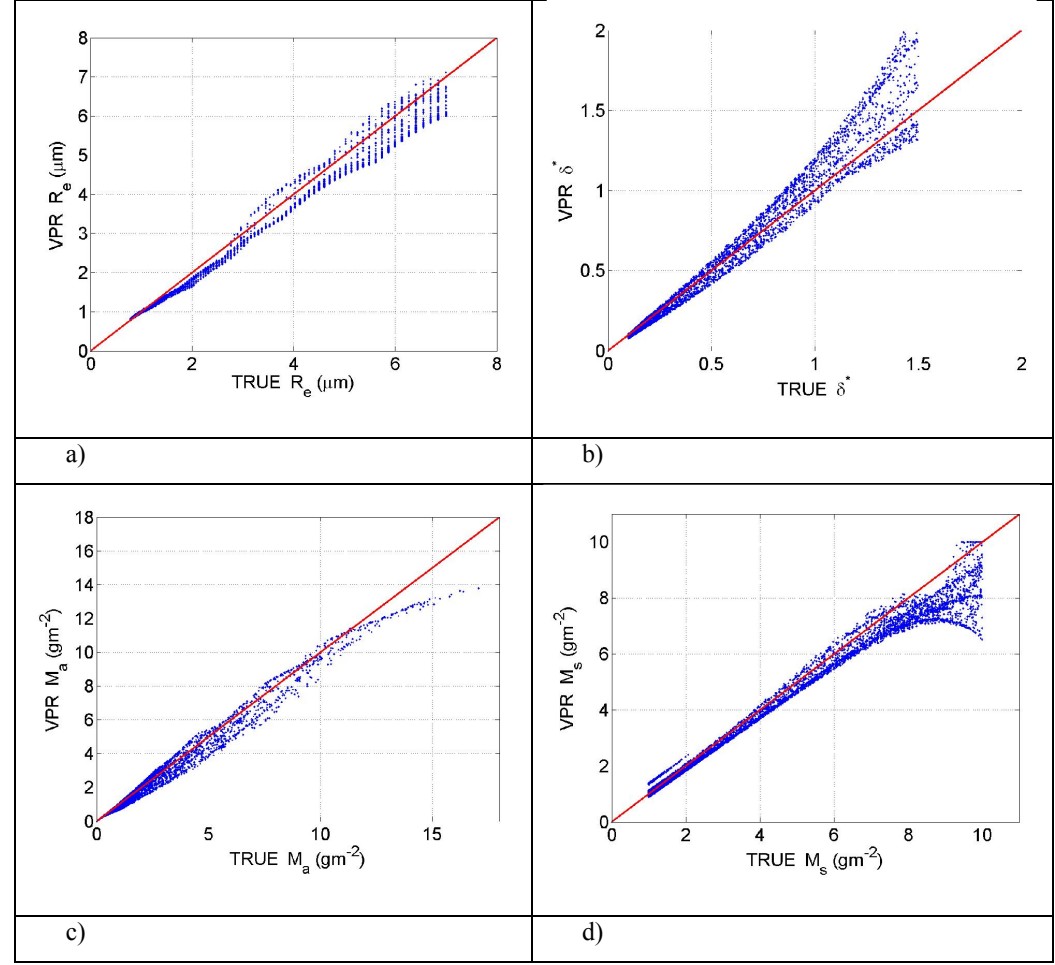

1 Figure 10. Eyjafjallajökull-Andesite example: scatter plots between VPR results and true values:

2 effective radius (a), ash optical depth at 550 nm (b), ash mass (c), and $SO_2$ mass (d). Red line is the

3 bisector.



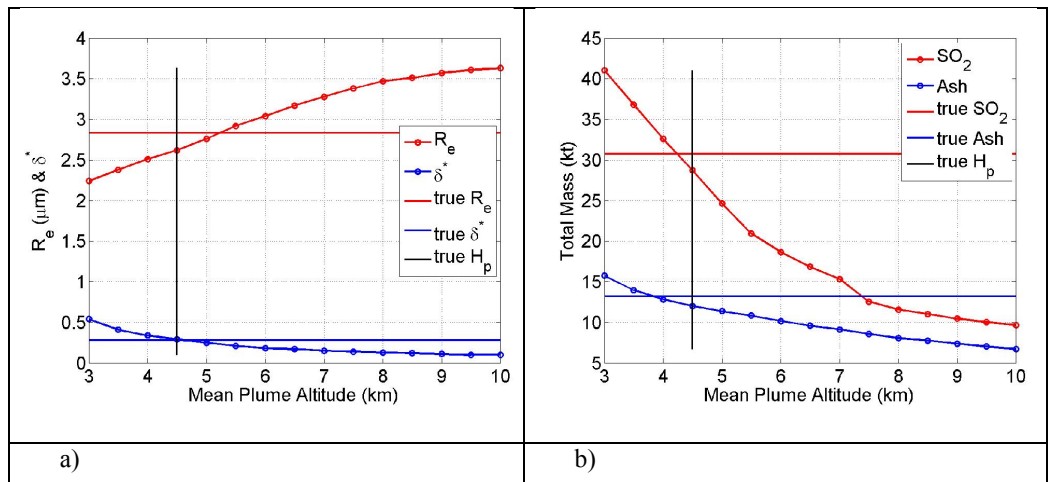

Figure 11. Eyjafjallajökull-Andesite example: trends of $R_e$ and $\delta^*$ mean values (a), and ash and $SO_2$ total mass (b) retrieved by VPR with different input plume altitudes.



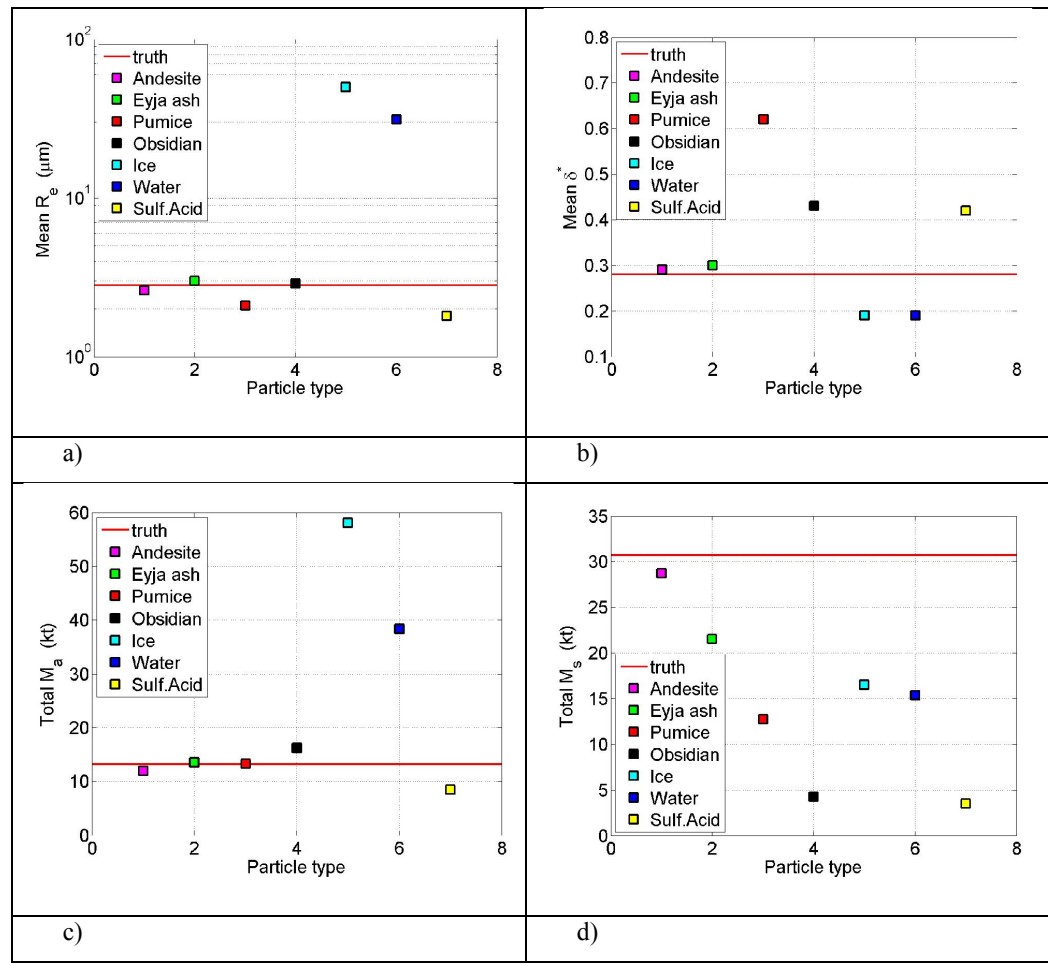

Figure 12. Eyjafjallajökull-Andesite example. VPR results with different types of particles: mean effective radius (a), mean optical depth at 550 nm (b), total mass (c), and total $SO_2$ mass (d). The red lines are the true values.