# Peer review of "Real time retrieval of volcanic cloud particles and SO2 by satellite using an improved simplified approach"

_Atmospheric Measurement Techniques, 2015_

## Referee Comment (RC1) · Anonymous Referee #2 · 10 Mar 2016

**Review of ACPD manuscript amt-2015-390: Real time retrieval of volcanic cloud particles and SO2 by satellite using an improved simplified approach**

This paper describes improvements of an existing procedure, Volcanic Plume Removal (VPR) that retrieves optical depth, effective radius, and mass of volcanic ash particles as well as Sulphur Dioxide (SO2) mass. The improved approach makes use of empirically established relationships between the Planck emission at the mean volcanic temperature ($B(T_p)$ ) and other volcanic plume properties in scenarios with and without SO2 presence using radiative transfer calculations. The topic of the paper is of scientific interest but, as currently written, the paper appears confusing and incomplete. Comments and suggestions for addressing these issues are given below.

Nowhere in the manuscript are the retrieval of ash particles properties and SO2 concentrations discussed. The title should not make reference to the actual retrieval of ash particles and SO2 concentrations which are not really addressed in the paper.

The paper is really about the use of radiative transfer calculations to develop linear relationships between the volcanic plume's Planck function at its mean temperature and empirically established fitting parameters ($B_{up}$ and $B_{dn}$) and between $B(T_p)$ and the volcanic plume transmittance ($\tau_t$) in the absence of sulfur dioxide. In the presence of SO2, linear relationships are established between $B(T_p)$ and a confusing constant $B_s$ that depends on multiple parameters (i.e., plume temperature; an undefined plume position, and also an undefined state of the atmosphere above the plume) and, therefore, not a constant by definition.

The connection between the above mentioned linear functions and the actual retrieval of ash particle properties and SO2 is not established. It is left up to the reader to figure out what the 'subsequent steps of the VPR procedure' are. As currently written the paper resembles more a technical document to be circulated between members of a closed research community who are familiar with the intricacies of the VPR, and not a document of interest to a larger community interested in general aspects of the ash retrieval problem but unfamiliar with the details of the VPR. My suggestion to the authors is to either narrow down the scope of the paper to the parametrization exercise mentioning its application to improve VPR, or expand the discussion part of the paper to describe succinctly but clearly the remaining aspects of the VPR with references to the previously published work.

Other comments:

What is the physical meaning of $\alpha$ in Equation 1? What are its units?

Terms $\tau^{'}$ and $\tau^{''}$ are not explicitly defined in the text.

An explanation for the transition from Equation 1 to Equations 2 and 3 is lacking. It appears that in arriving to the simplified expressions in Equation 2, the term $\alpha$ becomes zero, $\tau^{''}$ becomes unity, and the term $L^{``}_{uo}$ vanishes. A short explanation on the physical basis associated with the algebraic transformation should be added.

On page 6 it should be added coefficients *a* and *b* are, respectively, the slope and y–intercept of the linear fits illustrated in Fig. 4a.

The use of the term 'volcanic particles' is confusing.  Use either 'volcanic ash' or 'sulfate aerosols' in the appropriate context. Although they are both generated by the eruption, their properties, lifetimes, and importance are quite different.

---

## Referee Comment (RC2) · Anonymous Referee #3 · 11 Mar 2016

**"Real time retrieval of volcanic cloud particles and SO$_2$ by satellite using an improved simplified approach" by S. Pugnaghi et al.**

The manuscript presents Volcanic Plume Removal (VPR) developed for ash optical depth, effective radius and mass, and sulphur dioxide mass using an improved simplified approach. Unfortunately, the paper is difficult to understand for the general reader of Atmospheric Measurement Techniques (AMT). Therefore, I am not recommending the publication of this paper as it is.

1. Sometimes it is difficult to understand what the authors want to present in this manuscript. In particular, weak logical connections between paragraphs and sentences and too-long sentences to catch in Abstract and Introduction (e.g. page 1 lines 22-24, pages 2 lines 13-19) can make the reader confusing. Please rephrase and clarify the manuscript for general AMT readers. Additionally, please discuss more in detail about other satellite products comparable to your products using a new table summarizing the accuracy and characteristic.

2. The manuscript looks like a technical document for the experts. Please explain in more detail about the essential equations with clear and consistent definitions for the variables.

3. The authors need to enhance the validation part by spatial comparison against other satellite- and ground-based observations from other publications if possible. Furthermore, try to explain why they are different.

---

## Referee Comment (RC3) · Anonymous Referee #1 · 14 Mar 2016

**Review of the paper 'Real time retrieval of volcanic cloud particles and SO$_2$ by satellite using an improved simplified approach' by S. Pugnaghi et al.**

The manuscript describes an improvement to an earlier published Volcanic Plume Removal (VPR) procedure for retrieval of volcanic ash mass, ash particle effective radius and SO$_2$ mass. In its present form the manuscript more resembles a technical report than a full scientific paper. Some suggestions for improvement are given below.

**Major comments**

My main concern is with the rather limited scope of the synthetic data set used for testing of the VPR procedure. As it stands the synthetic data set is too limited and sparse to allow for an adequate testing and understanding of the strenghts and weaknesses of the improved VPR procedure. To address this at least the following two tests with accompanying discussions should be included:

- The improved procedure is tested on two idealized synthetic images. This is a very limited and not very instructive comparison for the general usefulness and applicability of the procedure. To demonstrate the real life behaviour of the procedure it should also be tested on real data and examples of such included. For example could the real meausurements (or similar) on which the synthetic images are based, be analysed.

- The synthetic data set only includes ash and SO$_2$. It is relatively easy to include water and ice clouds into such synthetic data sets and this has been done earlier by other authors. The authors are strongly urged to also include a synthetic test which include water and ice clouds together with ash and SO$_2$. Realistic ice and water clouds may be taken from for example ECMWF or similar forecast models. Inclusion of such test cases will greatly improve the testing space for the VPR procedure. This will also allow to include estimates of false positive and false negative ash and SO$_2$ pixels detection for the VPR procedure. Such estimates are extremely useful in order to fully comprehend the qualities of the VPR procedure.

**Minor comments**

- **Page 1, line 19**: Please move the parenthesis listing plume particles from lines 22-23 to line 19 and insert after "particles".

- **Page 1, line 25**: Include the years for the Mt. Etna and Eyjafjallajökull eruptions which the synthetic images resemble.

- **Page 2, line 32**: Give years of eruptions.

- **Page 4, line 10**: $L_p$ should not be in bold face.

- **Page 4, line 13**: Please explain $\tau'$, $\tau''$ and $L'_{u0}$ or point the reader to the Appendix for explanation.

- **Page 7, line 32**: Please also mention that details of synthetic image generation are included in the following two subsections 3.1 and 3.2.

- **Page 8, line 18-19**: Please include the ash mass loading that the optical depths of 0.1 and 1.5 translate into.

- **Page 8, line 19**: $\delta$ should not be in bold face.

- **Page 10, line 1-23**: Please discuss what approximations in the VPR procedure that are the main reasons for the differences seen between the ash and $SO_2$ clouds input to the calculation of the synthetic images and the corresponding retrieved values.

- **Page 12, line 8**: Should it not be $S = 0$?

- **Page 27, line 1**: Please write something like "Synthetic RGB images constructed from bands at 8.7, 11, and 12 $\mu$m " instead of "Synthetic images (radiance at the sensor); RGB: bands at 8.7, 11, and 12 m respectively".

---

## Author Comment (AC1) · 9 May 2016

Dear Editor,

we apologise: our paper had to be really confused if two of three Reviewers consider it unclear. We hope that the new version, revised according to the Reviewers suggestions, is clearer. We wish to thank all the Reviewers for their useful comments.

General Answers to the Reviewers

1. The main aspect of the proposed paper is the estimation of the transmittance of a volcanic cloud (at 8.7, 11 and 12 μm) directly from satellite images knowing only the temperature of the volcanic cloud assumed at a constant altitude or knowing the vertical profile of the air temperature.
   From the plume transmittances, with simple expressions already described in AMT (Pugnaghi et al., 2013), the optical depth, the effective radius, the mass of ash and sulphur dioxide contained in the volcanic cloud can be estimated.
   VPR needs a set of parameters. The paper describes how to compute them and reports their values obtained for: both MODIS sensors; some common ash types, ice crystals and water droplets; two important European volcanoes: Mt. Etna and Eyjafjallajökull.
   The submitted paper improves the main step of a simplified approach to perform the real time retrieval of volcanic cloud particle and $SO_2$ by satellite TIR images, as the title says. It also provides the required parameters for all the aforementioned cases.
2. VPR is an approximated procedure described and published on AMT (Pugnaghi et al., 2013), used in Annals of Geophysics (Corradini et al., 2014), adapted for SEVIRI sensor on JVGR (Guerrieri et al., 2015), used also on Remote Sensing (Corradini et al., 2016). During the inter-comparison exercise (SCOPE) presented in Madison, WI, (June-July 2015) and funded by WMO, the VPR results were closely compared with the results obtained with the main other available methodologies for volcanic ash cloud retrieval.
3. To verify the advantages of the improved model were proposed two synthetic images. In our opinion, this is one of the best methods because the volcanic cloud content is known. But the Reviewers have different opinions.

The new version has been simplified to make it more readable: the theoretical part (section 2) has been modified and the mathematical appendix A has been removed. The application of the method (section 3) has been expanded and split in three parts. The first describes the distributions of the differences between the input data used to perform the MODTRAN simulations and the VPR retrieval, exactly as described in AMT (Pugnaghi et al., 2013). The second part, as in the submitted version of this paper, shows the VPR retrievals using two synthetic images. The description has been simplified and the results reduced, but a case with a volcanic cloud containing ice crystals and sulphur dioxide has been added as required by one Reviewer (RC3). The last part shows the VPR retrievals of a real case: the Eyjafjallajökull eruption on May 11, 2010 and the comparison with the LUT (Corradini et al.,2009) retrievals for the same image as required by one Reviewer (RC3).

Finally, in the new version we have omitted the part related to the sulphuric droplets because of the reduced sensitivity of the method for effective radii greater than 8 microns.

**Specific answer to the RC1:**

*This paper describes improvements of an existing procedure, Volcanic Plume Removal (VPR) that retrieves optical depth, effective radius, and mass of volcanic ash particles as well as Sulphur Dioxide (SO2) mass. The improved approach makes use of empirically established relationships between the Planck emission at the mean volcanic temperature ($B(T_p)$ ) and other volcanic plume properties in scenarios with and without SO2 presence using radiative transfer calculations. The topic of the paper is of scientific interest but, as currently written, the paper appears confusing and incomplete. Comments and suggestions for addressing these issues are given below.*

*Nowhere in the manuscript are the retrieval of ash particles properties and SO2 concentrations discussed. The title should not make reference to the actual retrieval of ash particles and SO2 concentrations which are not really addressed in the paper.*

*The paper is really about the use of radiative transfer calculations to develop linear relationships between the volcanic plume's Planck function at its mean temperature and empirically established fitting parameters ($B_{up}$ and $B_{dn}$) and between $B(T_p)$ and the volcanic plume transmittance ($\tau_t$) in the absence of sulfur dioxide. In the presence of SO2, linear relationships are established between $B(T_p)$ and a confusing constant $B_s$ that depends on multiple parameters (i.e., plume temperature; an undefined plume position, and also an undefined state of the atmosphere above the plume) and, therefore, not a constant by definition.*

*The connection between the above mentioned linear functions and the actual retrieval of ash particle properties and SO2 is not established. It is left up to the reader to figure out what the 'subsequent steps of the VPR procedure' are. As currently written the paper resembles more a technical document to be circulated between members of a closed research community who are familiar with the intricacies of the VPR, and not a document of interest to a larger community interested in general aspects of the ash retrieval problem but unfamiliar with the details of the VPR. My suggestion to the authors is to either narrow down the scope of the paper to the parametrization exercise mentioning its application to improve VPR, or expand the discussion part of the paper to describe succinctly but clearly the remaining aspects of the VPR with references to the previously published work.*

As said above, the submitted paper describes how to get the relationship between the radiance at the sensor and the volcanic cloud transmittance. The method to calculate the optical thickness and effective radius of the particles contained in each pixel of the volcanic cloud, and then the mass, is described in a previous AMT paper. It is not repeated, because it is not changed and very easy to understand.

In the paper we show how to approximate the relationship between the radiance at the sensor and the transmittance of the volcanic cloud with two linear trends, one representing the transparent part of the cloud and one the most opaque. This is done for the three considered MODIS-TIR bands, affected by the presence of liquid or solid particle in the cloud. To determine the two mentioned linear trends is sufficient to know the temperature at the mean altitude of the volcanic cloud.

To determine the transmittance due to the sulphur dioxide at 8.7 μm only one linear trend is required (the gas does not scatter). $B_s$ is the offset of this linear trend and again it is obtained by knowing the temperature of the volcanic cloud at its mean altitude. $B_s$, as clearly says equation (8; 7 in the new version), is a constant for that volcanic cloud. Clearly, if the height of the cloud changes, it will change

also $B_s$ or, if the height is the same but will change the temperature at that altitude (that is, the state of the atmosphere changes) also $B_s$ will change and so on. For a specific eruption with the plume at a given altitude, $B_s$ is a constant.

*Other comments:*

*What is the physical meaning of α in Equation 1? What are its units?*

Alpha ($\alpha$) represents the radiance scattered along the line of sight of the sensor because of the presence of the volcanic cloud (W m$^{-2}$ sr$^{-1}$ μm$^{-1}$). It was said in the text (pag. 4. Line 15) and also in the appendix (pag. 13, line 6). The old equation (1) has been removed in the new version because it is not used in the model. Therefore the previous appendix A has been eliminated too, making the paper shorter and simpler.

*Terms $\tau'$ and $\tau''$ are not explicitly defined in the text.*

$\tau''$ is defined on pag. 4, line 6. It is right, $\tau'$ was not defined because practically not used. Anyway, on pag. 4, line 16 it is said that $\tau=\tau'*\tau''$ is the total transmittance of the atmosphere. Therefore, if $\tau''$ is the transmittance of the layer of atmosphere above the volcanic cloud, $\tau'$ is the transmittance of the layer of atmosphere below the volcanic cloud, as shown in Fig. 2. However, $\tau'$ will be defined in the new version too.

*An explanation for the transition from Equation 1 to Equations 2 and 3 is lacking. It appears that in arriving to the simplified expressions in Equation 2, the term α becomes zero, $\tau''$ becomes unity, and the term $L_{uo}''$ vanishes. A short explanation on the physical basis associated with the algebraic transformation should be added.*

Equation (1) says that to account for the radiance scattered towards the sensor by the particles of the volcanic cloud, a parabolic function is required, nothing else. As can be seen in Fig. 3a the parabolic function well explains the main part of the cases, but it is not so good for the very thick clouds and becomes useless for transparent plumes, where a linear trend is sufficient. Equation (2) and (3) are not algebraically derived from Equation (1). As can be seen in Fig. 3b, equation (2) simply approximates the part of parabola of equation (1) for higher transmittance values (transparent plume), while equation (3) well approximates the part of parabola obtained for lower transmittance values (opaque plume).

More precisely, equation (2) represents the cases with negligible $\alpha$, but not necessarily with $\tau''$ equal unit and null $L_{uo}''$, as supposed by the Reviewer. If it was so, then the coefficients ($a_{up}$ and $b_{up}$) in Tables S1 to S7 would always have been 1 and 0 respectively.

However, in the new version of the paper eq. (1) and Fig. 3a have been removed.

*On page 6 it should be added coefficients a and b are, respectively, the slope and y–intercept of the linear fits illustrated in Fig. 4a.*

That is right "a" and "b" are slope and offset of the linear fits shown in Fig. 4a, it has been added.

*The use of the term 'volcanic particles' is confusing. Use either 'volcanic ash' or 'sulfate aerosols' in the appropriate context. Although they are both generated by the eruption, their properties, lifetimes, and importance are quite different.*

RC1 is right, and the term 'volcanic particles' might be confusing. We had a short discussion about the term to use in the paper when speaking about the solid and liquid particles (aerosols) present in the volcanic cloud. Ash is a very common term, but when in addition to volcanic ash or sulphate aerosol the considered particle is or might be also a water droplet or a crystal of ice, we believe that the more inclusive term 'volcanic cloud particles' is also more precise.

**Specific answer to the RC2:**

*The manuscript presents Volcanic Plume Removal (VPR) developed for ash optical depth, effective radius and mass, and sulphur dioxide mass using an improved simplified approach. Unfortunately, the paper is difficult to understand for the general reader of Atmospheric Measurement Techniques (AMT). Therefore, I am not recommending the publication of this paper as it is.*

*1. Sometimes it is difficult to understand what the authors want to present in this manuscript. In particular, weak logical connections between paragraphs and sentences and too--‑long sentences to catch in Abstract and Introduction (e.g. page 1 lines 22--‑24, pages 2 lines 13--‑19) can make the reader confusing. Please rephrase and clarify the manuscript for general AMT readers. Additionally, please discuss more in detail about other satellite products comparable to your products using a new table summarizing the accuracy and characteristic.*

As said in the general comments, this paper shows how to estimate the volcanic cloud transmittance from the three quoted MODIS bands in the TIR atmospheric window without evaluating the atmospheric correction, and knowing only the air temperature at the mean altitude of the plume as additional input. This improves the previous VPR procedure, already presented and published on AMT, accounting for the scattering of the thermal radiance in the line of sight of the sensor. The paper also reports, for two important European volcanoes, the coefficients required by the procedure to estimate the plume transmittances for different types of ash and for crystals of ice and droplets of water.

This procedure is specific for radiometers with 8.7, 11, and 12 μm bands in the TIR atmospheric window, and the reported parameters are for the MODIS instrument on board Aqua and Terra satellites.

We hope that the new version of the paper will be more interesting and clear for the general AMT readers. Comparisons with other satellite products is out of the scope of this paper and has already done and presented in previous articles. Please read General answers to Reviewers.

*2. The manuscript looks like a technical document for the experts. Please explain in more detail about the essential equations with clear and consistent definitions for the variables.*

The current version of the paper has been simplified to make it more readable. The essential equations are the ones reported in the paper. Equation (1) of the first version has been omitted to prevent possible misunderstandings and to shorten the paper. Also he appendix A of the previous version has been removed because considered no more necessary. All the variables used in the equations are now clearly defined.

*3. The authors need to enhance the validation part by spatial comparison against other satellite--- and ground---based observations from other publications if possible. Furthermore, try to explain why they are different.*

The most important questions that this paper has to answer are: can this new model account for the plume scattering, and are its retrieval at least as accurate as the results obtained with the first VPR version? We think that the simplest way to answer to these questions is to use dummy images with volcanic clouds completely known. In the General answers to the Reviewers we remind that the VPR has been already compared with the other main procedures for volcanic ash and $SO_2$ cloud retrieval, and this paper does not aim to re-do that. However, results obtained analysing a real image are reported in the new version and compared with the retrieval based on the LUT procedure (Corradini et al.,2009) for the same MODIS image, having assumed the same volcanic cloud altitude. This comparison, like those carried out with the dummy images, reduces the possible ambiguities and/or differences and gives a good idea of the approximations introduced with this simplified procedure.

**Specific answer to the RC3:**

*The manuscript describes an improvement to an earlier published Volcanic Plume Removal (VPR) procedure for retrieval of volcanic ash mass, ash particle effective radius and SO2 mass. In its present form the manuscript more resembles a technical report than a full scientific paper. Some suggestions for improvement are given below.*

*Major comments*

*My main concern is with the rather limited scope of the synthetic data set used for testing of the VPR procedure. As it stands the synthetic data set is too limited and sparse to allow for an adequate testing and understanding of the strenghts and weaknesses of the improved VPR procedure. To address this at least the following two tests with accompanying discussions should be included:*

*The improved procedure is tested on two idealized synthetic images. This is a very limited and not very instructive comparison for the general use-fullness and applicability of the procedure. To demonstrate the real life behaviour of the procedure it should also be tested on real data and examples of such included. For example could the real measurements (or similar) on which the synthetic images are based, be analysed.*

*The synthetic data set only includes ash and SO2. It is relatively easy to include water and ice clouds into such synthetic data sets and this has been done earlier by other authors. The authors are strongly urged to also include a synthetic test which include water and ice clouds together with ash and SO2. Realistic ice and water clouds may be taken from for example ECMWF or similar forecast models. Inclusion of such test cases will greatly improve the testing space for the VPR procedure. This will also allow to include estimates of false positive and false negative ash and SO2 pixels detection for the VPR procedure. Such estimates are extremely useful in order to fully comprehend the qualities of the VPR procedure.*

The aim of the submitted paper is to show how calculate the transmittance of a volcanic cloud in the three common thermal infrared bands of the atmospheric window directly from the remotely sensed radiances. To do this, not only two simple synthetic images have been used, but more than 400 thousands simulations of the radiance at the sensor; this for each type of presented particles and for the two considered volcanoes. The knowledge of the cloud transmittance is very important from many scientific points of view.

The two synthetic images are used to give only a first idea of how the improved model works, respect to the previous VPR version. Nevertheless, the two used images are not "idealized images". They correspond to the two real atmospheres measured by MODIS close to the two chosen volcanoes during their actual eruptions, and the geometries of the two volcanic clouds used for the synthetic clouds are the real geometries of the clouds emitted during those true eruptions. The synthetic part concerns only the knowledge of the type and the amount of aerosol contained in the cloud, which means to know the 'truth' about the retrievals.

In the General answers to the Reviewers we remind that the VPR has been already compared with the other main procedures for volcanic ash and $SO_2$ cloud retrieval, and this paper does not aim to re-do that. However, results obtained analysing a real image are reported in the new version and compared with the retrieval based on the LUT procedure (Corradini et al.,2009) for the same MODIS image, having assumed the same volcanic cloud altitude. This comparison, like those carried out with the dummy images, reduces the possible ambiguities and/or differences and gives a good idea of the approximations introduced with this simplified procedure.

Moreover, in the new version of the paper the distributions of the differences between the input data used in the above-mentioned simulations and the results of the VPR have been added. They are compared with the distribution obtained from the previous version (published on AMT).

An extra synthetic image has been added too; it contains crystals of ice and $SO_2$. The effect of the ice is similar to the water droplets effect, but the ice seems more common than water in volcanic clouds. The added case is only for Mt. Etna because the parameters for the ice were not available in the older version of VPR for the Eyjafjallajökull volcano.

The equations described in the submitted paper give the volcanic cloud transmittance at 8.7, 11 and 12 μm knowing the radiance at the sensor at that wavelengths and the plume temperature. $SO_2$ affects the first band (8.7 μm) and it is used to compute the $SO_2$ abundance. The transmittances of the other two bands are used to compute the optical thickness and the effective radius of the particle contained in the considered pixel. With only two equations, it is possible to consider only one type of aerosol per pixel. Therefore, if ice and $SO_2$, and ash and $SO_2$ are present together in the same volcanic cloud, the retrieval is still possible if it is possible to discriminate the two parts with suitable masks. In other words, to run the VPR procedure two separate masks have to be defined and used, one for each type

of particle considered as prevalent in each pixel. Mixed pixel are not considered because there are not enough equations to carry out the retrievals with this method.

VPR does not perform the volcanic cloud detection, but it requires it. Volcanic plume, meteorological cloud, land-sea masks have to be considered all as input data (see Fig. 1). They are usually performed with separate and already known classification algorithms.

*Minor comments*

*Page 1, line 19: Please move the parenthesis listing plume particles from lines 22-23 to line 19 and insert after "particles".*

OK, done.

*Page 1, line 25: Include the years for the Mt. Etna and Eyjafjallajokull eruptions which the synthetic images resemble.*

This information was present in the section describing the synthetic images and now is in the new section 3.2.

*Page 2, line 32: Give years of eruptions.*

The quoted parameters are for all the possible eruptions in the two areas, not only for some specific case. The text has been modified.

*Page 4, line 10: Lp should not be in bold face.*

Equation (1) has been omitted and also this Lp

*Page 4, line 13: Please explain $\tau'$, $\tau''$ and L'uo or point the reader to the Appendix for explanation.*

Equation (1) and appendix have been removed. All the other symbols are described.

*Page 7, line 32: Please also mention that details of synthetic image generation are included in the following two subsections 3.1 and 3.2.*

This part has been changed, but the suggested mention has been included.

*Page 8, line 18-19: Please include the ash mass loading that the optical depths of 0.1 and 1.5 translate into.*

To simplify the description this detail has been neglected.

*Page 8, line 19: δ should not be in bold face.*

δ is no more bold.

*Page 10, line 1-23: Please discuss what approximations in the VPR procedure that are the main reasons for the differences seen between the ash and SO2 clouds input to the calculation of the synthetic images and the corresponding retrieved values.*

To simplify the reading and make the paper less a report, the indicated part has been removed. Nevertheless, the two previous synthetic images remained and a new one with ice and sulphur dioxide has been added. Also the distributions of the differences between the input data, used to perform all the MODTRAN simulations, and the VPR retrievals are now reported.

VPR is an approximated procedure to estimate the volcanic cloud transmittance from the measured radiance at the sensor. Its first step is the definition of a virtual image seen by the sensor, as if the volcanic plume were not present. This is done linearizing the radiance along all the plume transects. Then there is the most important step, the one described in this paper, that is the evaluation of the plume transmittances directly from the measured radiance. Also this derives from a linearization or better from a two steps linearization.

These are the main reasons producing differences between input and expected data. The linearization along the transects of the plume depends on how much the surface (and the atmosphere) on both sides of the plume is uniform. The used synthetic images are perfectly uniform. The linearization of the radiance at the sensor versus the plume transmittance depends mainly on the type of particle, on the radius of the used particle and on the optical thickness, but in minor quantity on the viewing angle and so on.

The two VPR versions are different only in the determination of the volcanic cloud transmittance. The first version does not consider the radiance scattered towards the sensor; it neglects the layer of atmosphere above the plume, but it adjusts it by means of a cubic fit. The new version takes into account the scattering of the thermal radiance to the sensor; it does not neglect the layer of atmosphere above the plume and does not make use of adjusting fits. These differences reflect their effects on the results; in some specific cases, the approximations performed with the older procedure can be better than the one performed with the new one, but it is not the general case.

*Page 12, line 8: Should it not be S = 0?*

In the new version, the appendix is omitted. Anyway, S term is zero only if the scattering is absent; that is, if no particles but only gases are present in the volcanic cloud.

*Page 27, line 1: Please write something like \Synthetic RGB images constructed from bands at 8.7, 11, and 12 m \ instead of \Synthetic images (radiance at the sensor); RGB: bands at 8.7, 11, and 12 m respectively".*

To simplify and shortening the paper, the RGB images have been removed.

**References:**

Corradini, S., Montopoli, M., Guerrieri, L., Ricci, M., Scollo, S., Merucci, L., Marzano, F. S., Pugnaghi, S., Prestifilippo, M., Ventress, L., Grainger, R. G., Carboni, E., Vulpiani, G., Coltelli, M.: A multi-sensor approach for the volcanic ash cloud retrieval and eruption characterization: the 23 November 2013 Etna lava fountain, Remote Sens. 2016, 8, 58; doi:10.3390/rs8010058.

Corradini, S., Pugnaghi, S., Piscini, A., Guerrieri, L., Merucci, L., Picchiani, M., Chini M.: Volcanic Ash and SO2 retrievals using synthetic MODIS TIR data: comparison between inversion procedures and sensitivity analysis, Annals of Geophysics, Special Issue on "Atmospheric Emissions from Volcanoes", Vol. 57, Fast Track 2, 2014. doi: 10.4401/ag-6616.

Corradini, S., Merucci, L., and Prata, A. J.: Retrieval of SO2 from thermal infrared satellite measurements: correction procedures for the effects of volcanic ash, Atmos. Meas. Tech., 2, 177–191, doi:10.5194/amt-2-177-2009, 2009

Guerrieri, L., Merucci, L., Corradini, S., and S. Pugnaghi, Evolution of the 2011 Mt. Etna ash and SO2 lava fountain episodes using SEVIRI data and VPR retrieval approach, Journal of Volcanology and Geothermal Research, Vol. 291, pp. 63-71, doi:10.1016/j.jvolgeores.2014.12.016, (2015).

Pugnaghi, S., Guerrieri, L., Corradini, S., Merucci, L., and Arvani, B.: A new simplified approach for simultaneous retrieval of SO2 and ash content of tropospheric volcanic clouds: an application to the Mt Etna volcano, Atmos. Meas. Tech., 6, 1315-1327, doi:10.5194/amt-6-1315-2013, 2013.